# GLOBAL EXPLAINABILITY OF GNNS VIA LOGIC COMBINATION OF LEARNED CONCEPTS

**Steve Azzolin**[1]**, Antonio Longa**[1,3]**, Pietro Barbiero**[2]**, Pietro Liò**[2] **& Andrea Passerini**[1]
[1]University of Trento, [2]University of Cambridge, [3]Fondazione Bruno Kessler
`{steve.azzolin}@studenti.unitn.it`, `{pb737, pl219}@cam.ak.uk`
`{antonio.longa, andrea.passerini}@unitn.it`

## ABSTRACT

While instance-level explanation of GNN is a well-studied problem with plenty of approaches being developed, providing a *global* explanation for the behaviour of a GNN is much less explored, despite its potential in interpretability and debugging. Existing solutions either simply list local explanations for a given class, or generate a synthetic prototypical graph with maximal score for a given class, completely missing any combinatorial aspect that the GNN could have learned. In this work, we propose GLGExplainer (Global Logic-based GNN Explainer), the first Global Explainer capable of generating explanations as arbitrary Boolean combinations of learned graphical concepts. GLGExplainer is a fully differentiable architecture that takes local explanations as inputs and combines them into a logic formula over graphical concepts, represented as clusters of local explanations. Contrary to existing solutions, GLGExplainer provides accurate and human-interpretable global explanations that are perfectly aligned with ground-truth explanations (on synthetic data) or match existing domain knowledge (on real-world data). Extracted formulas are faithful to the model predictions, to the point of providing insights into some occasionally incorrect rules learned by the model, making GLGExplainer a promising diagnostic tool for learned GNNs.

## 1 INTRODUCTION & RELATED WORK

Graph Neural Networks (GNNs) have become increasingly popular for predictive tasks on graph structured data. However, as many other deep learning models, their inner working remains a black box. The ability to understand the reason for a certain prediction represents a critical requirement for any decision-critical application, thus representing a big issue for the transition of such algorithms from benchmarks to real-world critical applications.

Over the last years, many works proposed Local Explainers (Ying et al., 2019; Luo et al., 2020; Yuan et al., 2021; Vu & Thai, 2020; Shan et al., 2021; Pope et al., 2019; Magister et al., 2021) to explain the decision process of a GNN in terms of factual explanations, often represented as subgraphs for each sample in the dataset. We leave to Yuan et al. (2022) a detailed overview about Local Explainers, who recently proposed a taxonomy to categorize the heterogeneity of those. Overall, Local Explainers shed light over *why* the network predicted a certain value for a specific input sample. However, they still lack a global understanding of the model. Global Explainers, on the other hand, are aimed at capturing the behaviour of the model as a whole, abstracting individual noisy local explanations in favor of a single robust overview of the model. Nonetheless, despite this potential in interpretability and debugging, little has been done in this direction. GLocalX (Setzu et al., 2021) is a general solution to produce global explanations of black-box models by hierarchically aggregating local explanations into global rules. This solution is however not readily applicable to GNNs as it requires local explanations to be expressed as logical rules. Yuan et al. Yuan et al. (2020) proposed XGNN, which frames the Global Explanation problem for GNNs as a form of input optimization (Wu et al., 2020), using policy gradient to generate synthetic prototypical graphs for each class. The approach requires prior domain knowledge, which is not always available, to drive the generation of valid prototypes. Additionally, it cannot identify any compositionality in the returned explanation, and has no principled way to generate alternative explanations for a given class. Indeed, our experi-

mental evaluation shows that XGNN fails to generate meaningful global explanations in all the tasks we investigated.

Concept-based Explainability (Kim et al., 2018; Ghorbani et al., 2019; Yeh et al., 2020) is a parallel line of research where explanations are constructed using "concepts" i.e., intermediate, high-level and semantically meaningful units of information commonly used by humans to explain their decisions. Concept Bottleneck Models (Koh et al., 2020) and Prototypical Part Networks (Chen et al., 2019a) are two popular architectures that leverage concept learning to learn explainable-by-design neural networks. In addition, similarly to Concept Bottleneck Models, Logic Explained Networks (LEN) (Ciravegna et al., 2021a) generate logic-based explanations for each class expressed in terms of a set of input concepts. Such concept-based classifiers improve human understanding as their input and output spaces consist of interpretable symbols (Wu et al., 2018; Ghorbani et al., 2019; Koh et al., 2020). Those approaches have been recently adapted to GNNs (Zhang et al., 2022; Georgiev et al., 2022; Magister et al., 2022). However, these solutions are not conceived for explaining already learned GNNs.

**Our contribution** consists in the first Global Explainer for GNNs which *i)* provides a Global Explanation in terms of logic formulas, extracted by combining in a fully differentiable manner graphical concepts derived from local explanations; *ii)* is faithful to the data domain, i.e., the logic formulas, being derived from local explanations, are intrinsically part of the input domain without requiring any prior knowledge. We validated our approach on both synthetic and real-world datasets, showing that our method is able to accurately summarize the behaviour of the model to explain in terms of concise logic formulas.

## 2 BACKGROUND

### 2.1 GRAPH NEURAL NETWORKS

Given a graph $\mathcal{G} = (\mathcal{V}, \mathcal{E})$ with adjacency matrix $A$ where $A_{ij} = 1$ if there exists an edge between nodes $i$ and $j$, and a node feature matrix $X \in \mathbb{R}^{|\mathcal{V}| \times r}$ where $X_i$ is the $r$-dimensional feature vector of node $i$, a GNN layer aggregates the node's neighborhood information into a $d$-dimensional refined representation $H \in \mathbb{R}^{|\mathcal{V}| \times d}$. The most common form of aggregation corresponds to the GCN (Kipf & Welling, 2016) architecture, defined by the following propagation rule:

$$H^{k+1} = \sigma(\tilde{D}^{-\frac{1}{2}} \tilde{A} \tilde{D}^{-\frac{1}{2}} H^k W^k) \tag{1}$$

where $\tilde{A} = A + \boldsymbol{I}$, $\tilde{D}$ is the degree matrix relative to $\tilde{A}$, $\sigma$ an activation function, and $W \in \mathbb{R}^{F \times F}$ is a layer-wise learned linear transformation. However, the form of Eq 1 is heavily dependent on the architecture and several variants have been proposed (Kipf & Welling, 2016; Veličković et al., 2017; Gilmer et al., 2017)

### 2.2 LOCAL EXPLAINABILITY FOR GNNS

Many works recently proposed Local Explainers to explain the behaviour of a GNN (Yuan et al., 2022). In this work, we will broadly refer to all of those whose output can be mapped to a subgraph of the input graph (Ying et al., 2019; Luo et al., 2020; Yuan et al., 2021; Vu & Thai, 2020; Shan et al., 2021; Pope et al., 2019). For the sake of generality, let $\text{LEXP}(f, \mathcal{G}) = \hat{\mathcal{G}}$ be the weighted graph obtained by applying the local explainer LEXP to generate a local explanation for the prediction of the GNN $f$ over the input graph $\mathcal{G}$, where each $\hat{A}_{ij}$ relative to $\hat{\mathcal{G}}$ is the likelihood of the edge $(i, j)$ being an important edge. By binarizing the output of the local explainer $\hat{\mathcal{G}}$ with threshold $\theta \in \mathbb{R}$ we achieve a set of connected components $\bar{\mathcal{G}}_i$ such that $\bigcup_i \bar{\mathcal{G}}_i \subseteq \hat{\mathcal{G}}$. For convenience, we will henceforth refer to each of these $\bar{\mathcal{G}}_i$ as local explanation.

## 3 PROPOSED METHOD

The key contribution of this paper is a novel Global Explainer for GNNs which allows to describe the behaviour of a trained GNN $f$ by providing logic formulas described in terms of human-understandable concepts (see Fig. 1). In the process, we use one of the available Local Explainers (Ying et al., 2019; Luo et al., 2020; Yuan et al., 2021; Vu & Thai, 2020; Shan et al., 2021; Pope

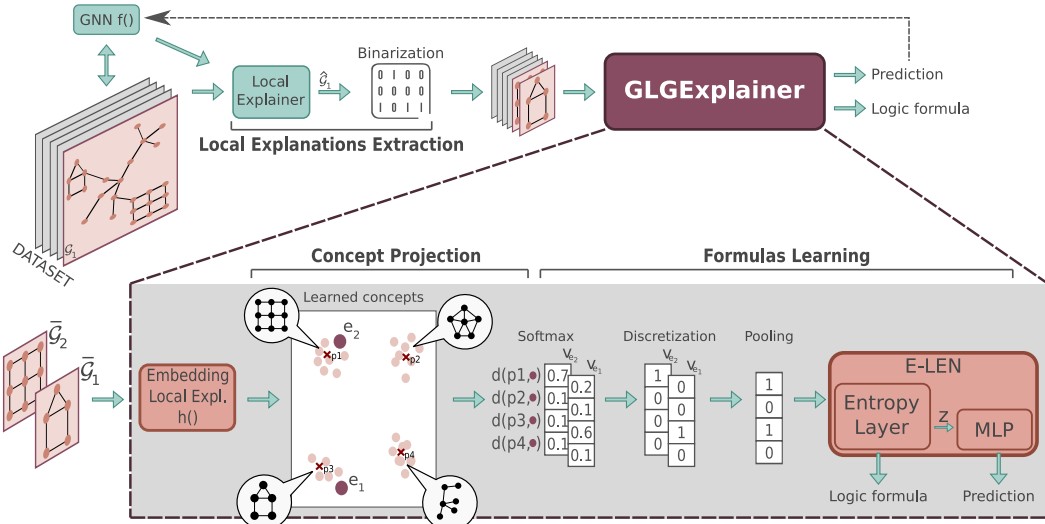

Figure 1: Illustration of the proposed method.

et al., 2019) to obtain a local explanation for each sample in the dataset. We then map those local explanations to some learned prototypes which will represent the final high-level concepts (e.g. motifs in a graph). Finally, for the formulas extraction, we input the vector of concept activations to an Entropy-based LEN (E-LEN) (Barbiero et al., 2021; Ciravegna et al., 2021b) which is trained to match the predictions of $f$. In the following, we will describe every step in more detail.

**Local Explanations Extraction:** The first step of our pipeline consists in extracting local explanations. In principle, every Local Explainer whose output can be mapped to a subgraph of the input sample is compatible with our pipeline (Ying et al., 2019; Luo et al., 2020; Yuan et al., 2021; Vu & Thai, 2020; Shan et al., 2021; Pope et al., 2019). Nonetheless, in this work, we relied on PGExplainer (Luo et al., 2020) since it allows the extraction of arbitrary disconnected motifs as explanations and it gave excellent results in our experiments. The result of this preprocessing step consists in a list $D$ of local explanations, which are provided as input to the GLGExplainer architecture. More details about the binarization are available in Section 4.2.

**Embedding Local Explanations:** The following step consists in learning an embedding for each local explanation that allows clustering together functionally similar local explanations. This is achieved with a standard GNN $h$ which maps any graph $\bar{\mathcal{G}}$ into a fixed-sized embedding $h(\bar{\mathcal{G}}) \in \mathbb{R}^d$. Since each local explanation $\bar{\mathcal{G}}$ is a subgraph of an input graph $\mathcal{G}$, in our experiments we used the original node features of the dataset. Note, however, that those features can be arbitrarily augmented to make the aggregation easier. The outcome of this aggregation consists in a set $E = \{h(\bar{\mathcal{G}}), \ \forall \bar{\mathcal{G}} \in D\}$ of graph embeddings.

**Concept Projection**: Inspired by previous works on prototype learning (Li et al., 2017; Chen et al., 2019b), we project each graph embedding $e \in E$ into a set $P$ of $m$ prototypes $\{p_i \in \mathbb{R}^d | i = 1, \ldots, m\}$ via the following distance function:

$$d(p_i, e) = softmax\left(log(\frac{\|e - p_1\|^2 + 1}{\|e - p_1\|^2 + \epsilon}), \ldots, log(\frac{\|e - p_m\|^2 + 1}{\|e - p_m\|^2 + \epsilon})\right)_i \quad (2)$$

Prototypes are initialized randomly from a uniform distribution and are learned along with the other parameters of the architecture. As training progresses, the prototypes will align as prototypical representations of every cluster of local explanations, which will represent the final groups of graphical concepts. The output of this projection is thus a set $V = \{v_e, \ \forall e \in E\}$ where $v_e = [d(p_1, e), .., d(p_m, e)]$ is a vector containing a probabilistic assignment of graph embedding $e$ (thus the local explanation that maps to it) to the $m$ concepts, and will be henceforth referred to as *concept vector*.

**Formulas Learning:** The final step consists of an E-LEN, i.e., a Logic Explainable Network (Ciravegna et al., 2021a) implemented with an Entropy Layer as first layer (Barbiero et al.,

2021). An E-LEN learns to map a concept activation vector to a class while encouraging a sparse use of concepts that allows to reliably extract Boolean formulas emulating the network behaviour. We train an E-LEN to emulate the behaviour of the GNN $f$ feeding it with the graphical concepts extracted from the local explanations. Given a set of local explanations $\bar{\mathcal{G}}_1 \dots \bar{\mathcal{G}}_{n_i}$ for an input graph $\mathcal{G}_i$ and the corresponding set of the concept vectors $v_1 \dots v_{n_i}$, we aggregate the concept vectors via a pooling operator and feed the resulting aggregated concept vector to the E-LEN, providing $f(\mathcal{G}_i)$ as supervision. In our experiments we used a max-pooling operator. Thus, the Entropy Layer learns a mapping from the pooled concept vector to (i) the embeddings $z$ (as any linear layer) which will be used by the successive MLP for matching the predictions of $f$. (ii) a truth table $T$ explaining how the network leveraged concepts to make predictions for the target class. Since the input pooled concept vector will constitute the premise in the truth table $T$, a desirable property to improve human readability is discreteness, which we achieve using the Straight-Through (ST) trick used for discrete Gumbel-Softmax Estimator (Jang et al., 2016). In practice, we compute the forward pass discretizing each $v_i$ via *argmax*, then, in the backward pass to favor the flow of informative gradient we use its continuous version.

**Supervision Loss:** GLGExplainer is trained end-to-end with the following loss:

$$L = L_{surr} + \lambda_1 L_{R1} + \lambda_2 L_{R2} \tag{3}$$

where $L_{surr}$ corresponds to a Focal BCELoss (Lin et al., 2017) between the prediction of our E-LEN and the predictions to explain, while $L_{R1}$ and $L_{R2}$ are respectively aimed to push every prototype $p_j$ to be close to at least one local explanation and to push each local explanation to be close to at least one prototype (Li et al., 2017). The losses are defined as follows:

$$L_{surr} = -y(1-p)^\gamma \log p - (1-y)p^\gamma \log(1-p) \tag{4}$$

$$L_{R1} = \frac{1}{m} \sum_{j=1}^{m} \min_{\bar{\mathcal{G}} \in D} \|p_j - h(\bar{\mathcal{G}})\|^2 \tag{5}$$

$$L_{R2} = \frac{1}{|D|} \sum_{\bar{\mathcal{G}} \in D} \min_{j \in [1,m]} \|p_j - h(\bar{\mathcal{G}})\|^2 \tag{6}$$

where $p$ and $\gamma$ represent respectively the probability for positive class prediction and the *focusing* parameter which controls how much to penalize hard examples.

## 4 EXPERIMENTS

We conducted an experimental evaluation on synthetic and real-world datasets aimed at answering the following research questions:

- **Q1: Can GLGExplainer extract meaningful Global Explanations?**
- **Q2: Can GLGExplainer extract faithful Global Explanations?**

The source code of GLGExplainer, including the extraction of local explanations, as well as the datasets and all the code for reproducing the results is made freely available online[1].

### 4.1 DATASETS

We tested our proposed approach on three datasets, namely:

**BAMultiShapes:** BAMultiShapes is a newly introduced extension of some popular synthetic benchmarks (Ying et al., 2019) aimed to assess the ability of a Global Explainer to deal with logical combinations of concepts. In particular, we created a dataset composed of 1,000 Barabási-Albert (BA) graphs with attached in random positions the following network motifs: house, grid, wheel. Class 0 contains plain BA graphs and BA graphs enriched with a house, a grid, a wheel, or the three motifs together. Class 1 contains BA graphs enriched with a house and a grid, a house and a wheel, or a wheel and a grid.

---

[1] `https://github.com/steveazzolin/gnn_logic_global_expl`

**Mutagenicity:** The Mutagenicity dataset is a collection of 4,337 molecule graphs where each graph is labelled as either having a mutagenic effect or not. Based on Debnath et al. (1991), the mutagenicity of a molecule is correlated with the presence of electron-attracting elements conjugated with nitro groups (e.g. *NO2*). Moreover, compounds with three or more fused rings tend to be more mutagenic than those with one or two. Most previous works on this dataset have focused on finding the functional group *NO2*, since most of them struggle in finding compounds with more than two carbon rings.

**Hospital Interaction Network (HIN)**: HIN is a new real-world benchmark proposed in this work. It represents the third-order ego graphs for doctors and nurses in a face-to-face interaction network collected in an hospital (Vanhems et al., 2013). There are four types of individuals in the network: doctors (D), nurses (N), patients (P), and administrators (A). Such typologies constitute the feature vector for each node, represented as one-hot encoding. Each ego network is an instance, and the task is to classify whether the ego in the ego network is a doctor or a nurse (without knowing its node features, which are masked). More details about the dataset construction are available in the Appendix.

For Mutagenicity we replicated the setting in the PGExplainer paper (Luo et al., 2020), while for BAMultiShapes and HIN we trained until convergence a 3-layer GCN. Details about the training of the networks and their accuracies are in the Appendix.

## 4.2 IMPLEMENTATION DETAILS

**Local Explanations Extraction:** As discussed in Section 3, we used PGExplainer (Luo et al., 2020) as the Local Explainer. However, we modified the procedure for discretizing weighted graphs into a set of disconnected motifs. Indeed, the authors in Luo et al. (2020) limited their analysis to graphs that contained the ground truth motifs and proposed to keep the top-k edges as a rule-of-thumb for visualization purposes. For Mutagenicity, over which PGExplainer was originally evaluated, we simply selected the threshold $\theta$ that maximises the F1 score of the local explainer over all graphs, including those that do not contain the ground-truth motif. For the novel datasets BAMultiShapes and HIN, we adopted a dynamic algorithm to select $\theta$ that does not require any prior knowledge about the ground truth motifs. This algorithm resembles the elbow-method, i.e., for each local explanation sort weights in decreasing order and chooses as threshold the first weight that differs by at least 40% from the previous one. We believe that threshold selection for Local Explainers is a fundamental problem to make local explainers actionable, but it is often left behind in favor of top-*k* selections where *k* is chosen based on the ground-truth motif. In the Appendix we show some examples for each dataset along with their extracted explanations.

**GLGExplainer:** We implemented the Local Explanation Embedder $h$ as a 2-layers GIN (Xu et al., 2018) network for each dataset except HIN, for which we found a GATV2 (Brody et al., 2021) to provide better performance as the attention mechanism allows the network to account for the relative importance of the type of neighboring individuals. All layers consist of 20 hidden units followed by a non-linear combination of max, mean, and sum graph pooling. We set the number of prototypes $m$ to 6, 2, and 4 for BAMultiShapes, Mutagenicity, and HIN respectively (see Section 4.4 for an analysis showing how these numbers were inferred), keeping the dimensionality $d$ to 10. We trained using ADAM optimizer with early stopping and with a learning rate of $1e^{-3}$ for the embedding and prototype learning components and a learning rate of $5e^{-4}$ for the E-LEN. The batch size was set to 128, the focusing parameter $\gamma$ to 2, while the auxiliary loss coefficients $\lambda_1$ and $\lambda_2$ were set respectively to 0.09 and 0.00099. The E-LEN consists of an input Entropy Layer ($\mathbb{R}^m \rightarrow \mathbb{R}^{10}$), a hidden layer ($\mathbb{R}^{10} \rightarrow \mathbb{R}^5$), and an output layer with LeakyReLU activation function. We turned off the attention mechanism encouraging a sparse use of concepts from the E-LEN, as the end-to-end architecture of GLGExplainer already promotes the emergence of predictive concepts, and the discretization step preceding the E-LEN encourages each concept activation vector to collapse on a single concept. All these hyper-parameters were identified via cross-validation over the training set.

## 4.3 EVALUATION METRICS

In order to show the robustness of our proposed methodology, we have evaluated GLGExplainer on the following three metrics, namely: *i)* FIDELITY, which represents the accuracy of the E-LEN in matching the predictions of the model $f$ to explain; *ii)* ACCURACY, which represents the accuracy

Table 1: Raw formulas as extracted by the Entropy Layer along with their test Accuracy. Each formula was rewritten to just keep positive literals.

| Dataset | | Raw Formulas | Accuracy |
|---|---|---|---|
| BAMultiShapes | $\text{Class}_0 \iff$ | $P_0 \vee P_3 \vee P_1 \vee P_4 \vee P_5$ | 0.98 |
| | $\text{Class}_1 \iff$ | $(P_4 \wedge P_3) \vee (P_5 \wedge P_4) \vee (P_3 \wedge P_1) \vee (P_5 \wedge P_1) \vee$ $(P_4 \wedge P_1) \vee (P_4 \wedge P_2) \vee (P_1 \wedge P_2) \vee (P_3 \wedge P_2) \vee$ $P_2$ | |
| Mutagenicity | $\text{Class}_0 \iff$ | $P_1 \vee (P_0 \wedge P_1)$ | 0.83 |
| | $\text{Class}_1 \iff$ | $P_0$ | |
| HIN | $\text{Class}_0 \iff$ | $P_0 \vee P_1 \vee P_3$ | 0.84 |
| | $\text{Class}_1 \iff$ | $P_2$ | |

of the formulas in matching the ground-truth labels of the graphs; *iii)* CONCEPT PURITY, which is computed for every cluster independently and measures how good the embedding is at clustering the local explanations. A more detailed description of those metrics is available in the Appendix. For BAMultiShapes and Mutagenicity, every local explanation was annotated with its corresponding ground-truth motif, or *Others* in case it did not match any ground-truth motif. For HIN, since no ground truth explanation is available, we labeled each local explanation with a string summarizing the users involved in the interaction (e.g., in a local explanation representing the interaction between a nurse and a patient, the label corresponds to *NP*). Since labelling each possible combination of interactions between 4 types of users would make the interpretation of the embedding difficult, we annotated the most frequent seven (*D, N, NA, P, NP, MN, MP*), assigning the rest to *Others*. Note that such unsupervised annotation may negatively impact the resulting Concept Purity since it ignores the correlation between similar but not identical local explanations.

## 4.4 EXPERIMENTAL RESULTS

In this section we will go through the experimental results with the aim of answering the research questions defined above. Table 1 presents the raw formulas extracted by the Entropy Layer where, to improve readability, we have dropped negated literals from clauses. This means that for each clause in a formula, missing concepts are implicitly negated. Those formulas can be further rewritten in a more human-understandable format after inspecting the representative elements of each cluster as shown in Figure 2, where for each prototype $p_j$ the local explanation $\bar{\mathcal{G}}$ such that $\bar{\mathcal{G}} = argmax_{\bar{\mathcal{G}}' \in D} d(p_j, h(\bar{\mathcal{G}}'))$ is reported. The resulting Global Explanations are reported in Figure 3, where we included a qualitative comparison with the global explanations generated by XGNN (Yuan et al., 2020), the only available competitor for global explanations of GNNs. Finally, Table 2 shows the results over the three metrics as described in Section 4.3. Note that XGNN is not shown in the table as it cannot be evaluated according to these metrics.

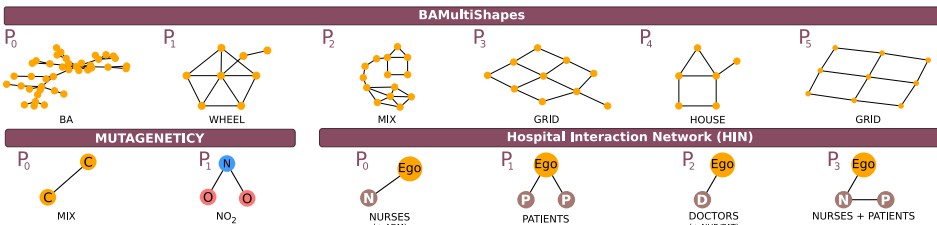

Figure 2: A representative element for each concept. For completeness, in the Appendix we report five random instances for each concept.

**Q1: Can GLGExplainer extract meaningful Global Explanations?**
The building blocks of the Global Explanations extracted by GLGExplainer are the graphical concepts that are learned in the concept projection layer. Figure 2 clearly shows that each concept rep-

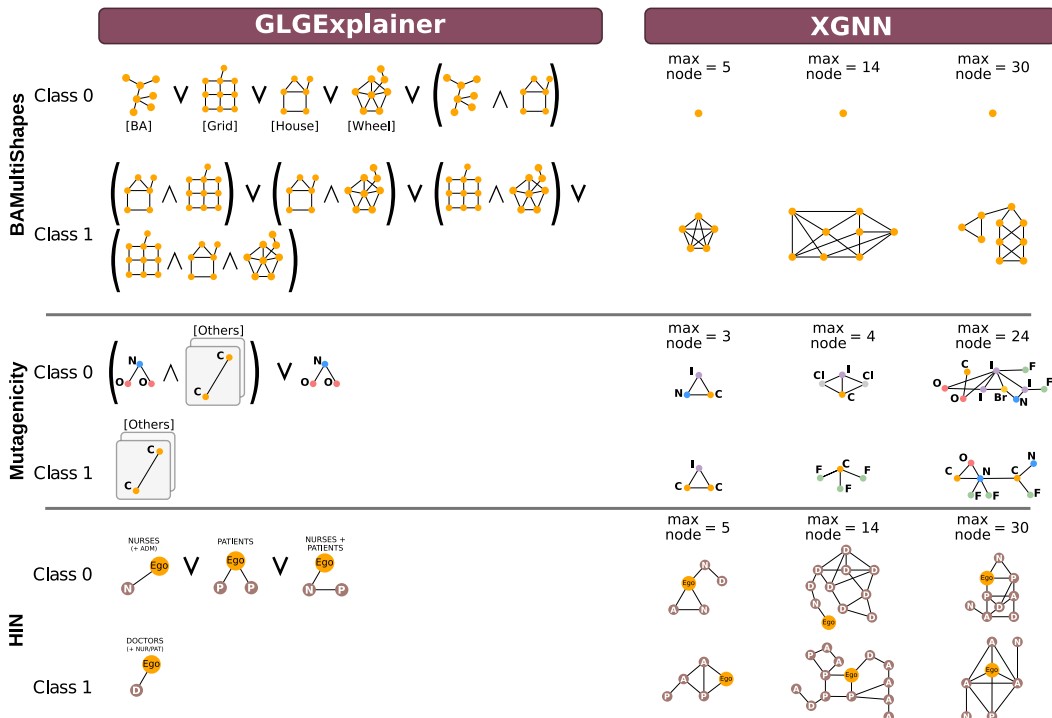

Figure 3: Global explanations of GLGExplainer (ours) and XGNN. For Class 0 of BAMultiShapes, XGNN was not able to generate a graph with confidence $\geq 0.5$. Note that for each clause, missing concepts are implicitly negated.

Table 2: Mean and standard deviation for Fidelity, Accuracy, and Concept Purity computed over 5 runs with different random seeds. Since the Concept Purity is computed for every cluster independently, here we report mean and standard deviation across clusters over the best run according to the validation set.

| Dataset | Fidelity | | Accuracy | | Test Concept |
| | Train | Test | Train | Test | Purity |
| --- | --- | --- | --- | --- | --- |
| BAMultiShapes | $0.96 \pm 0.03$ | $0.96 \pm 0.03$ | $0.92 \pm 0.03$ | $0.96 \pm 0.03$ | $0.87 \pm 0.24$ |
| Mutagenicity | $0.82 \pm 0.00$ | $0.81 \pm 0.01$ | $0.78 \pm 0.00$ | $0.79 \pm 0.01$ | $1.00 \pm 0.00$ |
| HIN | $0.89 \pm 0.00$ | $0.85 \pm 0.02$ | $0.86 \pm 0.01$ | $0.85 \pm 0.02$ | $0.78 \pm 0.18$ |

resents local explanations with specific characteristics, thus achieving the desired goal of creating interpretable concepts. Note that clusters corresponding to concepts are on average quite homogeneous (see Concept Purity in Table 2), and the concept representatives in the figure are faithful representations of the instances in their corresponding cluster. See the Appendix for further details, where we report five random instances for each concept. It is worth noting that this clustering emerges solely based on the supervision defined by Eq 3, while no specific supervision was added to cluster local explanations based on their similarity. This is the reason behind the emergence of the *Mix* cluster around $p_2$ in the upper part of Figure 2, which represents all local explanations with at least two motifs that are present solely in *Class 1* of BAMultiShapes.

Additionally, as shown in Figure 3, GLGExplainer manages to combine these building blocks into highly interpretable explanations. The explanation for BAMultiShapes almost perfectly matches the ground-truth formula, where the only difference is the conjunction of all motifs being assigned to *Class 1* rather than *Class 0*. This however is due to a discrepancy between the ground-truth formula and what the GNN learned, as will be discussed in the answer to **Q2**. Note that the *Mix* cluster has been rewritten as the conjunction of the shapes it contains when extracting the human-interpretable

formulas. For Mutagenesis, the GLGExplainer manages to recover the well-known NO2 motif as an indicator of mutagenicity (*Class 0*). It is worth reminding that in all formulas in Figure 3, negative literals have been dropped from clauses for improved readability. Having formulas for Mutagenesis only two concepts, this implies that the formula for *Class 0* actually represents NO2 itself ((NO2 ∧ OTHERS) ∨ (NO2 ∧ ¬ OTHERS) ⟺ NO2). For HIN, the global explanations match the common belief that nurses tend to interact more with other nurses or with patients, while doctors tend to interact more frequently with other doctors (or patients, but less frequently than nurses).

Conversely to our results, XGNN (Yuan et al., 2020) fails to generate explanations matching either the ground truth or common belief about the dataset in most cases, while failing to generate any graph in others.

**Q2: Can GLGExplainer extract faithful Global Explanations?**
The high Accuracy reported in Table 2 shows that the extracted formulas are correctly matching the behaviour of the model in most samples, while being defined over fairly pure concepts as shown by the Concept Purity in the same Table. It is worth highlighting that GLGExplainer is not simply generating an explanation for the ground-truth labeling of the dataset, bypassing the GNN it is supposed to explain, but it is indeed capturing its underlying predictive behaviour. This can be seen by observing that over the training set, Fidelity is higher than Accuracy for all datasets. Note that on BAMultiShapes, training accuracy is lower than test accuracy. The reason for this train-test discrepancy can be found in the fact that the GNN fails to identify the logical composition of all three motifs (which are rare and never occur in the test set) as an indicator of *Class 0*. This can be seen by decomposing the GNN accuracy with respect to the motifs that occur in the data (Table 3), and observing that the accuracy for the group with all three motifs (All) is exactly zero. Quite remarkably, GLGExplainer manages to capture this anomaly in the GNN, as the clause involving all three motifs is learned as part of the formula for *Class 1* instead of *Class 0*, as shown in Figure 3. The ability of GLGExplainer to explain this anomalous behaviour is a promising indication of its potential as a diagnostic tool for learned GNNs.

Table 3: Accuracy of the model to explain on the train set of BAMultiShapes with respect to every combination of motifs to be added to the BA base graph. *H, G, W* stand respectively for House, Grid, and Wheel.

| | Class 0 | | | | | Class 1 | | |
| --- | --- | --- | --- | --- | --- | --- | --- | --- |
| **Motifs** | ∅ | H | G | W | All | H + G | H + W | G + W |
| **Accuracy** (%) | 1.0 | 1.0 | 0.85 | 1.0 | 0.0 | 1.0 | 0.98 | 1.0 |

In the rest of this section we show how the number of prototypes, that critically affects the interpretability of the explanation, can be easily inferred by trading-off Fidelity, Concept Purity and sparsity, and we provide an ablation study to demonstrate the importance of the Discretization trick.

**Choice of the number of prototypes**: The number of prototypes in the experiments of this section was determined by selecting the smallest $m$ which achieves satisfactory results in terms of Fidelity and Concept Purity, as measured on a validation set. Specifically, we aim for a parsimonious value of $m$ to comply to the human cognitive bias of favoring simpler explanations to more complex ones (Miller, 1956). Table 4 reports how different values of $m$ impact Fidelity and Concept Purity. The number of prototypes achieving the best trade-off between the different objectives was identified as 6, 2 and 4 for BAMultiShapes, Mutagenicity, and HIN respectively.

**Role of the Discretization trick**: The Discretization trick was introduced in Section 3 to enforce a discrete prototype assignment, something essential for an unambiguous definition of the concepts on which the formulas are based. We ran an ablation study to evaluate its contribution to the overall performance of GLGExplainer. Figure 4 (left) shows the reduction in concept vector entropy achieved by GLGExplainer with the discretization trick (red curve, zero entropy by construction) as compared to GLGExplainer with the trick disabled (orange curve). Figure 4 (middle) reports the Fidelity over the training epochs for the two variants. The figure shows the effectiveness of the discretization trick in boosting Fidelity of the extracted formulas, which is more than double the one achieved disabling it. We conjecture that the reason for this behaviour is the fact that the discretization trick forces the hidden layers of the E-LEN to focus on the information relative to the closest

Table 4: Fidelity and Concept Purity as functions of the number $m$ of prototypes in use. Results are referred to the validation set.

| Metric | Dataset | $m = 2$ | $m = 4$ | $m = 6$ | $m = 8$ |
|--------|---------|---------|---------|---------|---------|
| Fidelity | BAMultiShapes | 0.93 | 0.93 | 0.95 | 0.95 |
| | Mutagenicity | 0.83 | 0.83 | 0.84 | 0.79 |
| | HIN | 0.88 | 0.89 | 0.89 | 0.88 |
| Concept Purity | BAMultiShapes | 0.42 | 0.73 | 0.84 | 0.91 |
| | Mutagenicity | 0.97 | 0.99 | 0.96 | 0.99 |
| | HIN | 0.45 | 0.80 | 0.77 | 0.70 |

prototype, ignoring other positional information of local explanations. Thus, the E-LEN predictions are much better aligned with the discrete formulas being extracted, and indeed the Accuracy of the formulas matches the Fidelity of the E-LEN, which is shown in the right plot. On the other hand, GLGExplainer without discretization has a high Fidelity but fails to extract highly faithful formulas. Note that simply adding an entropy loss over the concept vector to the overall loss (Eq. 3) fails to achieve the same performance obtained with the discretization trick.

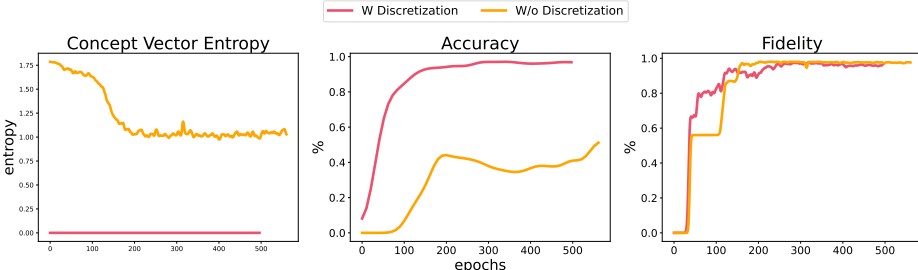

Figure 4: Impact of the Discretization Trick. Each run is halted with early stopping on the validation set.

## 5 CONCLUSION

We introduced GLGExplainer, the first Global Explainer for GNNs capable of generating explanations as logic formulas, represented in terms of learned human-interpretable graphical concepts. The approach is inherently faithful to the data domain, since it processes local explanations as extracted by an off-the-shelf Local Explainer. Our experiments showed that GLGExplainer, contrary to existing solutions, can faithfully describe the predictive behaviour of the model, being able to aggregate local explanations into meaningful high-level concepts and combine them into formulas achieving high Fidelity. GLGExplainer even managed to provide insights into some occasionally incorrect rules learned by the model. We believe that this approach can constitute the basis for investigating how GNNs build their predictions and debug them, which could substantially increase human trust in this technology.

The proposed GLGExplainer is inherently faithful to the data domain since it processes local explanations provided by a Local Explainer. However, the quality of those local explanations, in terms of representativeness and discriminability with respect to the task-specific class, has a direct effect on the Fidelity. If the generated concept vector does not exhibit any class-specific pattern, then the E-LEN will not be able to emulate the predictions of the model to explain. Despite being a potential limitation of GLGExplainer, this can actually open to the possibility of using the Fidelity as a proxy of local explanations quality, which is notoriously difficult to assess. We leave this investigation to future work. Despite tailoring our discussion on graph classification, our approach can be readily extended to any kind of classification task on graphs, provided that a suitable Local Explainer is available.

ACKNOWLEDGMENTS

This research was partially supported by TAILOR, a project funded by EU Horizon 2020 research and innovation programme under GA No 952215.

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

## A  APPENDIX

### A.1  DATASETS DETAILS

While BAMultiShapes and Mutagenicity were already described in detail in Section 4.1, here we report more details about the newly introduced dataset for graph Explainability. Figure 7-9 presents some random examples for each dataset, with their extracted explanation in bold.

**HIN:** Hospital Interaction Networks is a new real benchmark that we propose in this work. The dataset was collected using wearable sensors, equipped with radio-frequency identification devices (RFIDs) capturing face-to-face interactions. The devices record an interaction if and only if there is at least one exchanged signal within 20 seconds. The dataset was collected by Sociopatterns collaboration[2] in the geriatric ward of a university hospital (Vanhems et al., 2013) in Lyon, France, over four days in December 2010. The individuals belong to four categories: medical doctors (M), nurses (N), administrative staff (A), and patients (P).

Since the interaction network $\mathcal{GT}$ evolves over time, we convert the temporal network into a sequence of graph snapshots, aggregating interactions every five minutes ($\mathcal{GT} = [\mathcal{G}_1, \mathcal{G}_2, \ldots, \mathcal{G}_n]$). For each static graph $\mathcal{G}_i$, we extract the ego graph with radius 3 centered in each doctor and each nurse, obtaining a set of ego graphs. A GNN is trained in classifying between ego networks of doctor and nurse, where the feature of each ego node is masked. An illustrative example of this procedure is shown in Figure 5. In particular, the top of the figure shows a static snapshot ($\mathcal{G}_i$) of the temporal graph. In the middle, we show three ego graphs (with a radius equal to 2 for convenience) centered on two doctors ($D_1$ and $D_2$) and a nurse ($N_1$) respectively. Finally, the ego node features are masked, as depicted at the bottom of Figure 5. At the end of this extraction procedure, we obtain 880 and 2009 ego graphs for doctors and nurses respectively. To avoid class imbalance, we sub-sample the nurse class, obtaining a balanced dataset of 1760 graphs.

### A.2  IMPLEMENTATION DETAILS

#### A.2.1  TRAINING THE GNN $f$

In this section we will provide more details about the training of the GNN to explain $f$. While for Mutagenicity we limited to reproduce the results presented in (Luo et al., 2020) both in terms of

---

[2]http://www.sociopatterns.org/

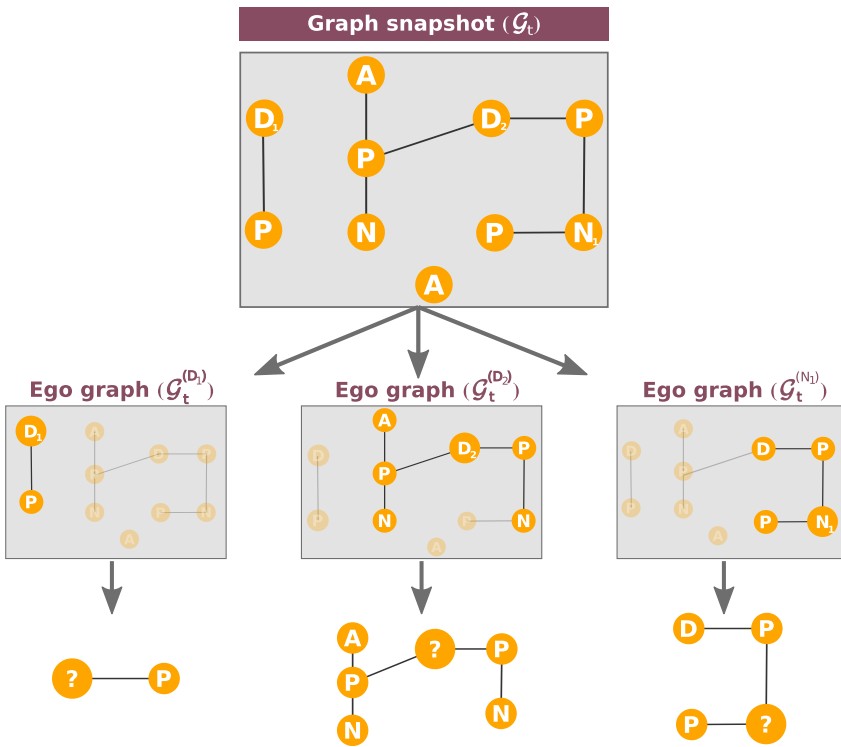

Figure 5: Extraction of ego graphs in Hospital Interaction Network (HIN).

model to explain (a 3 layers GCN Kipf & Welling (2016)) and local explanations, for BAMulti-Shapes and HIN we trained our own networks. For BAMultiShapes we adopted a 3-layers GCN (20-20-20 hidden units) with mean graph pooling for the final prediction, whilst for HIN we employed a 3-layers GCN (20-20-20 hidden units) with non-linear combination of sum, mean, and max graph pooling. The final model performances are reported in Table 5. In both cases we used ADAM optimizer, training until convergence and using the validation set to select the best epoch.

### A.2.2 EXPLAINERS

In this work we relied mainly on two off-the-shelf explainers, namely, PGExplainer (Luo et al., 2020) and XGNN (Yuan et al., 2020). Here we report some details about their usage.

**PGExplainer:** For Mutagenicity and BAMultiShapes, we used the original implementation as provided by Luo et al. (2020). For BAMultiShapes we changed to original hyper-parameters to {`epochs:5, args.elr=0.007, args.coff_t0=1.0, args.coff_size=0.0005 args.coff_ent=0.000`}. For HIN we instead used the PyTorch implementation provided by Agarwal et al. (2022), using as custom hyper-parameters {`t0=1, t1=1, max_epochs=30`}. Finally, for BAMultiShapes and HIN, for which we extracted our own local explanations, we trained PGExplainer on the train split of the original dataset. It is worth mentioning that for HIN, after the local explanations extraction, we removed every local explanation not containing the ego node.

**XGNN:** The official code provided here[3] is specifically tailored for generating explanations for the Mutagenicity dataset. For the other two, despite the algorithm accepting an heavy optimization for the task at hand (like defining custom rewards functions for each task), we made minimal changes to the architecture and to the original hyper-parameters in order not to input any a-priori knowledge. For HIN, specifically, we adapted the node type generation to match the node types of the dataset, and the custom `check_validity` function to define whether the generated graph is valid, i.e., it must contain a single ego node. For what concern the evaluation metrics presented in Section

---

[3]`https://github.com/divelab/DIG/tree/main/dig/xgraph/XGNN`

Table 5: Accuracies of the different models to explain.

| Split | BAMultiShapes | Mutagenicity | HIN |
|-------|---------------|--------------|------|
| Train | 0.94 | 0.87 | 0.92 |
| Val | 0.94 | 0.86 | 0.87 |
| Test | 0.99 | 0.86 | 0.86 |

4.3, since XGNN and GLGExplainer return explanations in two substantially different formats, we could not compare quantitatively the explanations provided by XGNN with ours. Thus, a direct comparison of XGNN under our metric is not possible.

### A.2.3 CHOOSING $\lambda_1$ AND $\lambda_2$

In Eq 3 we introduced the two parameters regulating the importance of the two auxiliary losses described in Section 3. Those parameters were kept fixed to, respectively, 0.09 and 0.00099 for all the experiments and were chosen via cross validation. In Figure 6 we show how the Fidelity over the validation set changes with different combinations of such hyper-parameters.

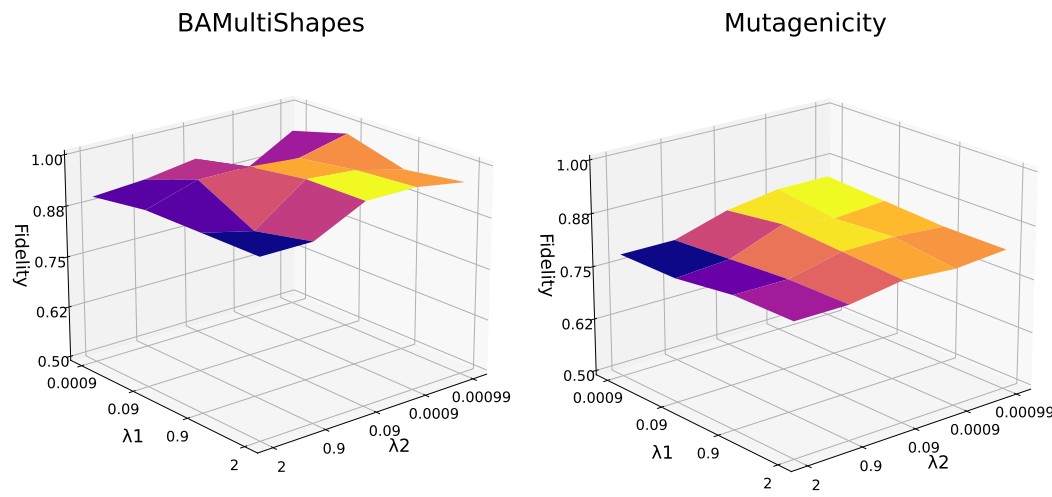

Figure 6: How does the selection of $\lambda_1$ and $\lambda_2$ impact Fidelity?

### A.3 LOCAL EXPLANATIONS EMBEDDING

For layout reasons, we did not report in the main paper the 2D embedding plot for each dataset. However, we believe it is of great interest since it gives a visual sense of how similar local explanations get clustered in the same concept. Thus, we report in Figure 10 the 2D plot of local explanations embedding for BAMultiShapes, Mutagenicity, and HIN.

In addition to this, we report a graphical materialization of five random samples for each concept, in Figures 11-13.

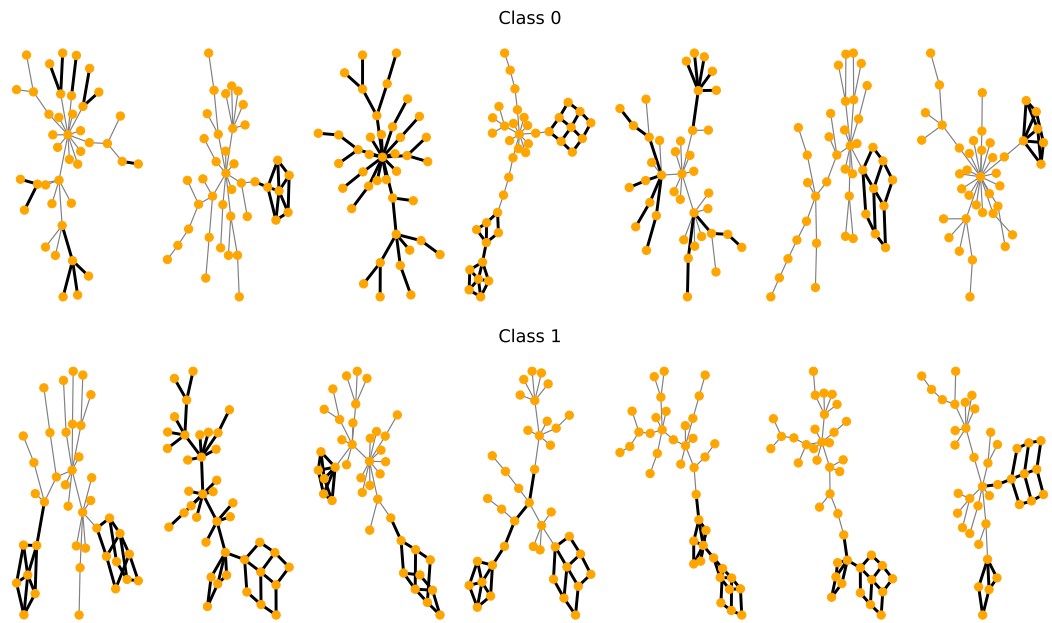

Figure 7: Random examples of input graphs along with their explanations in bold as extracted by PGExplainer, for BAMultiShapes.

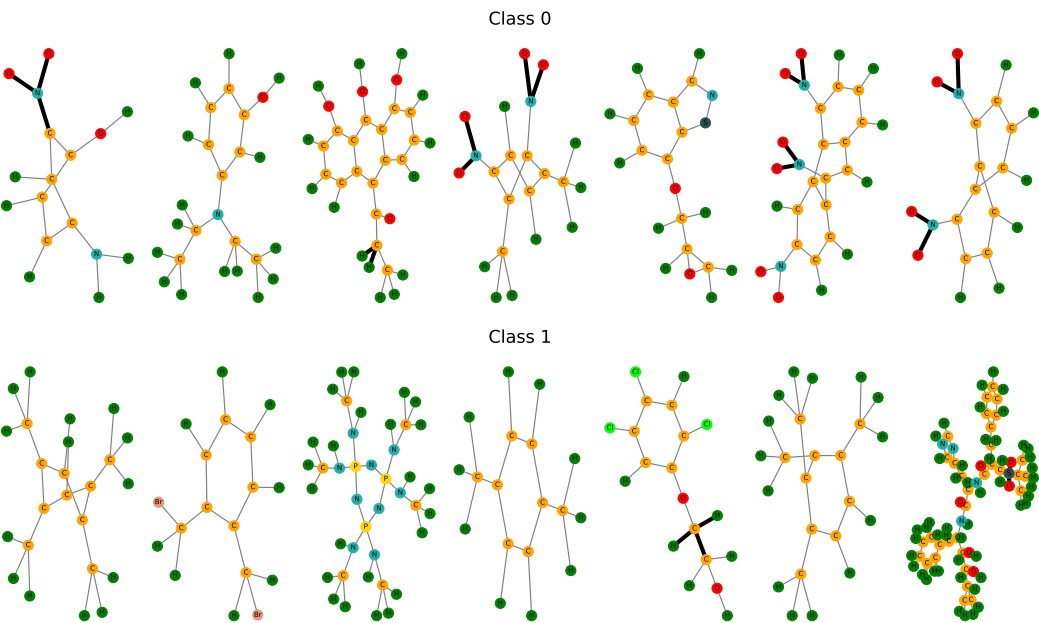

Figure 8: Random examples of input graphs along with their explanations in bold as extracted by PGExplainer, for Mutagenicity

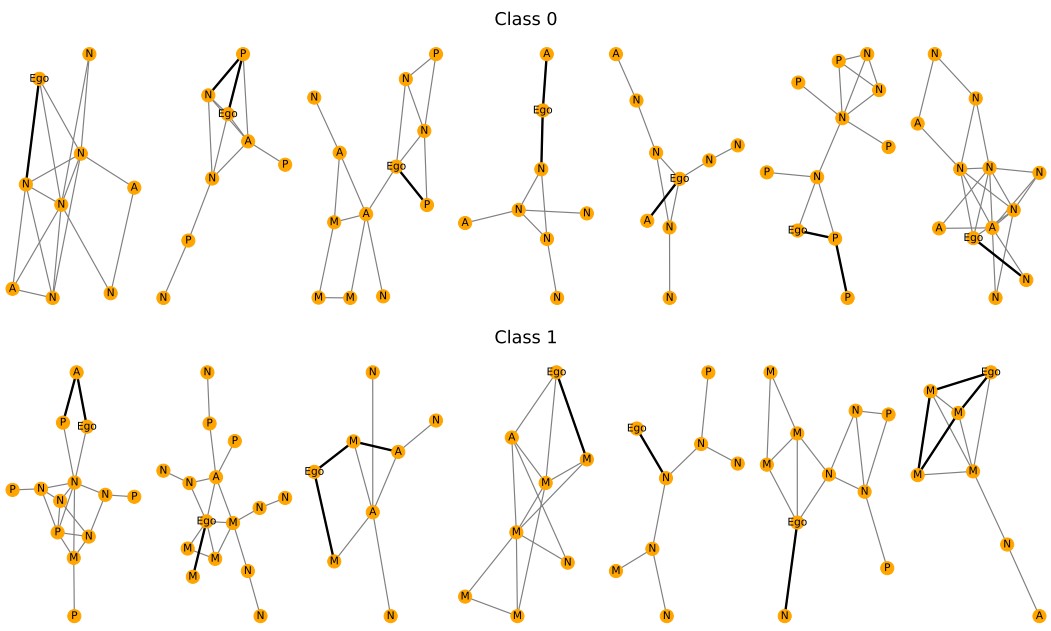

Figure 9: Random examples of input graphs along with their explanations in bold as extracted by PGExplainer, for HIN

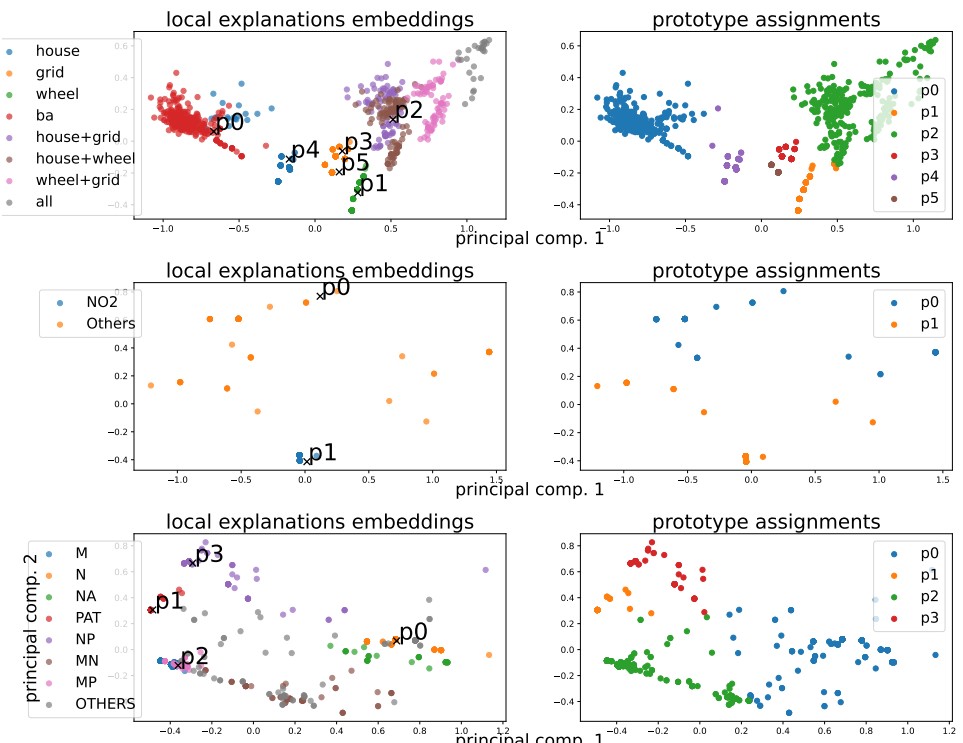

Figure 10: 2D PCA-reduced embedding for, respectively, BAMultiShapes and Mutagenicity and HIN

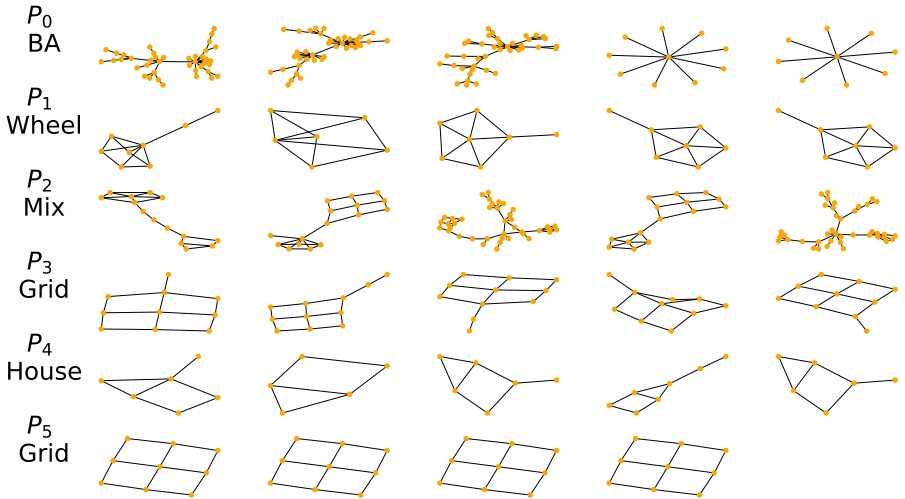

Figure 11: Five random local explanations for each concept in BAMultiShapes.

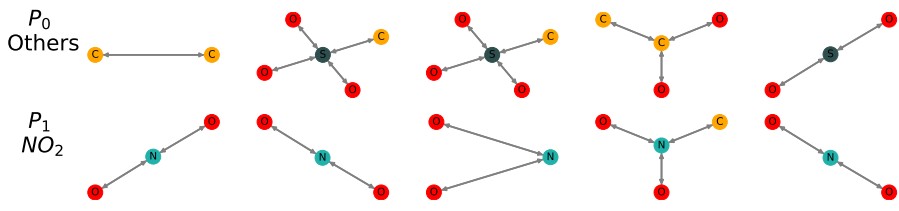

Figure 12: Five random local explanations for each concept in Mutagenicity.

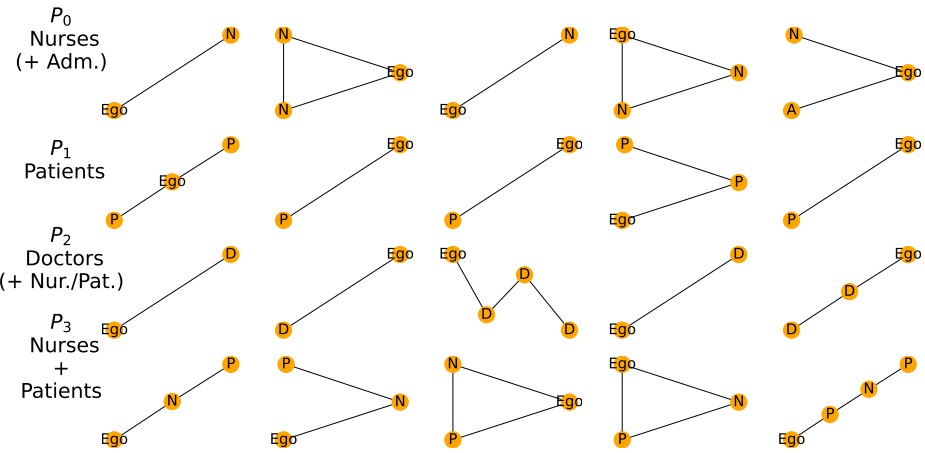

Figure 13: Five random local explanations for each concept in HIN.

## A.4 EVALUATION METRICS

Here we will describe in more detail the metrics briefly introduced in Section 4.3:

- Fidelity measures the accuracy between the prediction of the E-LEN and the one of the GNN to explain. It is computed as the accuracy between the class predictions of the E-LEN and the GNN $f$.

- Accuracy Barbiero et al. (2021) represents how well the learned formulas can correctly predict the class labels. To compute this metric, we treat the final formulas as a classifier that given an input concept vector predicts the class corresponding to the clause evaluated to true. In the cases in which either no clause or more clauses of different classes are evaluated to be true, the sample is always considered as wrongly predicted.

- Concept Purity is computed for every cluster independently and measures how good the embedding is at clustering the local explanations. It was first proposed in Magister et al. (2021) for evaluating concept representations by means of graph edit distance. However, since the computation of the graph edit distance is expensive, in our work we adapted such metric to exploit the annotation of local explanations as described in Section 4.3. Specifically, in our cases such annotation corresponds to the typology of the motif represented by the local explanation. The computation of the metric can be summarized by:

$$ConceptPurity(C_i) = \frac{count\_most\_frequent\_label(C_i)}{|C_i|} \tag{7}$$

where $C_i$ corresponds to the cluster having $p_i$ as prototype (i.e., the cluster containing every local explanation associated to prototype $p_i$ by the distance function $d(.,.)$ described in Section 3). $count\_most\_frequent\_label(C_i)$ instead returns the number of local explanations annotated with the most present label in cluster $C_i$. The Concept Purity results reported in Table 2 are computed by taking the mean and the standard deviation across the $m$ clusters.

## A.5 ENTROPY-BASED LOGIC EXPLAINED NETWORKS

Concept-based classifiers (Koh et al., 2020) are a family of machine learning models predicting class memberships from the activation scores of $k$ human-understandable categories, i.e., $q : \mathcal{C} \mapsto \mathcal{Y}$, where $\mathcal{C} \subseteq [0, 1]^k$. Concept-based classifiers improve human understanding as their input and output spaces consist of interpretable symbols (Wu et al., 2018; Ghorbani et al., 2019; Koh et al., 2020). Logic Explained Networks (LENs (Ciravegna et al., 2021a)) are concept-based neural models providing for each class $i$ simple logic explanations $\phi^i : \bar{\mathcal{C}} \mapsto \{0, 1\}$ for their predictions $\bar{q}^i(\bar{c}) \in \{0, 1\}$. In particular LENs provide concept-based First-Order Logic (FOL) explanations for each classification task:

$$\forall \bar{c} \in \bar{C} \subseteq \bar{\mathcal{C}} : \ \phi^i(\bar{c}) \leftrightarrow q^i(\bar{c}). \tag{8}$$

where $\phi^i$ is a concept-based formula in disjunctive normal form. The E-LEN employed in our work corresponds to a LEN with an Entropy Layer Barbiero et al. (2021) as first layer, which is the responsible for the extraction of the logic formulas. As mentioned in Section 4.2, given that our architecture already promotes by construction a singleton activation of concepts, the entropy-based regularization described in Barbiero et al. (2021) promoting a parsimonious activation of the concepts, allowing the E-LEN to predict class memberships using few relevant concepts only, is removed.

### A.5.1 EXTRACTION OF LOGIC FORMULAS

Considering the removal of the entropy-based regularization mentioned above, the process of formula extraction can be summarized as follows: given a classification task with $r$ classes, and given a truth table $T^i$ for each of the $r$ classes, the $l$-th row of the resulting table $T^i$ is obtained by concatenating together the $l$-th input concept activation vector $\bar{c}_l$ with the respective prediction $q^i(\bar{c}_l)$:

$$T_l^i = (\bar{c}_l \| q^i(\bar{c}_l)) \tag{9}$$

Then, for every row $l$, where $q^i(\bar{c}_l) = 1$, concepts in $\bar{c}_l$ are connected with the AND operator resulting in a logic clause where concepts that appear as false in $\bar{c}_l$ are negated. Finally, the final formulas in disjunctive normal form for table $T^i$ are obtained by connecting every clause with the OR operator. Further details are available in Barbiero et al. (2021).

### A.5.2 ENHANCEMENTS OF GLGEXPLAINER TO THE E-LEN FRAMEWORK

To make clear the enhancements of our proposed GLGExplainer to the E-LEN framework, note that the original formulation of the E-LEN requires $C$ to be known. Indeed, the experiments carried out in Barbiero et al. (2021) were all assuming a known mapping from the input space to $C$. In this case, however, since such mapping is not available, we aim at learning a set of human-understandable concepts. The choice of learning such concepts from local explanations, rather than generating them via any graph-generation process, allowed our contribution to be faithful to the data domain contrary to previous works on Global Explainability for GNNs Yuan et al. (2020). Since the E-LEN framework does not provide a principled way for dealing with the peculiarities of the graph domain, it cannot be directly applied on top of local explanations. We thus devised the method proposed in Section 3 in order to aggregate local explanations into clusters of similar subgraphs, with no further supervision than the one provided by Eq 3. The experimental evaluation in Section 4.4 shows that the learned clusters have indeed a human-understandable meaning, and the final formulas correctly allow to assess the performances of the model to explain. On the same vein, the framework proposed in Barbiero et al. (2021) is aimed at creating an interpretable classifier which can generate logic explanations of its predictions, resulting in an *interpretable-by-design* architecture. This is radically different from our scenario, where we provide explanations of an already trained model. Finally, given the fact that we removed the entropy-based regularization from the original implementation of the Entropy Layer[4], we are basically restricting the usage of it to just the principled process of extracting logic formulas from the truth table $T$, introduced in Barbiero et al. (2021).

### A.6 DISTANCE FUNCTION $d(.,.)$

In Section 3 we presented Eq. 2 as the distance function $d(p_i, e)$ to compute a relative assignment value for the graph embedding $e$ to prototype $p_i$, expressed as a probability value thanks to the softmax. For convenience, we report again the mathematical definition below:

$$d(p_i, e) = softmax \left( log(\frac{\|e - p_1\|^2 + 1}{\|e - p_1\|^2 + \epsilon}), \ldots, log(\frac{\|e - p_m\|^2 + 1}{\|e - p_m\|^2 + \epsilon}) \right)_i \qquad (10)$$

Note that when $d(p_i, e)$ has a high activation value for a prototype $p_i$, then the graph embedding $e$ is very close to prototype $p_i$ in the embedding space, thus meaning that the input graph leading to $e$ exhibit a similar high-level concept to what $p_i$ represents.

Overall, the output of the projection step described in Section 3 is a set $V = \{v_e, \ \forall e \in E\}$ where $v_e = [d(p_1, e), .., d(p_m, e)]$ is a vector containing a probabilistic assignment of graph embedding $e$ to the $m$ prototypes.

---

[4]https://pypi.org/project/torch-explain/

