# OpenReview forum: "Global Explainability of GNNs via Logic Combination of Learned Concepts"
_ICLR.cc/2023/Conference — ICLR 2023 poster_

### Official Review · Reviewer_8iKd · 2022-10-21

**Confidence:** 2
**Correctness:** 3
**Technical Novelty And Significance:** 2
**Empirical Novelty And Significance:** 2
**Recommendation:** 5

**Clarity, Quality, Novelty And Reproducibility:**

Language-wise and quality-wise the paper is well-written. The experimental results are described up to a sufficient level of detail. The experimental setup and the tool are unavailable and are promised to be released upon paper acceptance. As mentioned above, the novelty of the paper seems limited while the proposed approach represents a recipe applying the known and existing technology with no clear rationale provided.

**Strength And Weaknesses:**

First of all, I am not an expert on GNNs and their explainability and so I am afraid I am not the right person to judge the merits of this concrete work. For the same reason, I find this paper quite hard to follow.

This is a purely practical paper that describes how one can aggregate local explanations, provided in the form of sub-graphs of the input graphs, into a joint propositional formula that summarizes the behavior of the GNN model from the global perspective.

One of the upsides of the paper is that it is rigorous in terms of citations and seems to cover the prior work in the area quite extensively. It offers a large number of nicely drawn images illustrating the general flow and also examples in the experimental results, which is also a plus as they help a reader a lot.

Having said that, I have to mention that the proposed solution does not look like a sufficient contribution to me. Granted that this paper hinges on a significant amount of engineering work but the insight is a bit shallow. The fact that the approach is described in essentially a single page seems to confirm this issue. What is worse, the authors give no rationale behind the choices they propose to make. It is just a sequence of steps (1), (2), ..., followed by (n) and a non-expert reader like me is left wondering why these steps are better than the possible alternatives (if any). The steps themselves can be seen as a bunch of technicalities applied widely in the ML community and so the novelty seems a bit on the edge here (again, for a non-expert).

Experimental results are thoroughly presented and discuss the accuracy and fidelity of the computed explanations. However, these concepts do not look sufficient for measuring the "quality" of explanations, especially from the perspective of logic and formal correctness of those. I believe in this case a user study is necessary to confirm to what extend the explanations offered by this approach actually help a human understand the behavior of the model.

**Summary Of The Paper:**

This practical paper proposes a method for computing global explanations for the predictions made by graph neural networks (GNNs). The approach hinges on the use of a local GNN explainer invoked for computing local explanations for the GNN's predictions in the form of sub-graphs of the inputs. Given a number of local explanations, the proposed approach then combines them into a single propositional formula serving as a global summary of the GNN's behavior. The experimental results obtained on three datasets demonstrate the faithfulness and accuracy as well as high interpretability of the proposed approach.

**Summary Of The Review:**

In my view, the paper is under-delivering on the novelty side and represents purely practical/engineering effort. I doubt that this should meet the quality standards of a flagship AI/ML conference like ICLR. However, given the lack of my expertise, I admit that I may overlook some important merits of the work, hence my score is not entirely negative.

---

### Official Review · Reviewer_CpLz · 2022-10-23

**Confidence:** 3
**Correctness:** 3
**Technical Novelty And Significance:** 3
**Empirical Novelty And Significance:** 3
**Recommendation:** 8

**Clarity, Quality, Novelty And Reproducibility:**

- Generally, Clarity, Quality and Novelty of the paper is good.
- Probably there will be better reproducibility when the source codes are available.


**Strength And Weaknesses:**

- Strength
    - The paper is well written and organized, and readers can easily follow the content.
    - The paper organically aggregates a series of established methods, such as PGExplainer for extracting local explanation subgraphs, and concept or prototype based learning for human-understandable graphical concepts, and E-LEN for generating explanations with compositionality in form of Boolean combinations, in order to address an interesting and useful problem of capturing the behavior of the model as a whole.
- Weaknesses
    - There is only single baseline XGNN in the experiments. This is understandable, after all, the previous work on Global Explainer is limited. The table 2 only shows the performance of GLGExplainer, but does not show that of the baseline.
    - Although this paper mainly aggregates a series of established methods, it would be helpful if some specific details were explained more clearly in self-contained way.
        - For example, the Equation 2 and E-LEN


**Summary Of The Paper:**

This paper proposes GLGExplainer (Global Logic-based GNN Explainer), which is aimed at capturing the behavior of the model as a whole, abstracting individual noisy local explanations in favor of a single robust overview of the GNN model by generating explanations as arbitrary Boolean combinations of learned human-understandable graphical concepts. GLGExplainer uses PGExplainer to extract local explanations (as subgraphs) from graphs and learns an embedding for each local explanation that allows to cluster together functionally similar local explanations, and then projects each local explanation embedding into a set of m prototypes, which are initialized randomly from a uniform distribution and are learned along with the other parameters. The projection for each embedding is thus a vector containing a probabilistic assignment of local explanation embedding. Finally, a Logic Explainable Network learns to map a concept activation vector to a class while encouraging a sparse use of concepts that allows to reliably extract Boolean formulas emulating the network behavior. The authors validated GLGExplainer on both synthetic and real-world datasets, and claims that GLGExplainer is able to accurately summarize the behavior of the model to explain in terms of
concise logic formulas.

**Summary Of The Review:**

GLGExplainer is the first Global Explainer for GNNs in terms of logic formulas, and it offers a helpful and human-understandable diagnostic tool for learned blackbox GNNs.

---

### Official Review · Reviewer_gQT1 · 2022-10-25

**Confidence:** 4
**Correctness:** 3
**Technical Novelty And Significance:** 2
**Empirical Novelty And Significance:** 3
**Recommendation:** 5

**Clarity, Quality, Novelty And Reproducibility:**

The quality of the paper is fine except a lack of comparison with XGNN in terms of main metrics and some presentation problems. The clarity is fair considering that two important questions mentioned above have not been answered in the paper. The originality is marginal since the main idea is merely an application of the E-LEN framework to generate global explanations for GNNs. There is no code or data provided in the supplemental material.

**Strength And Weaknesses:**

Strengths:

(1) The work may have a significant impact since generating global explanations for GNNs is seldom studied by now. There seems only one notable approach to generating global explanations for GNNs, namely XGNN (Yuan et al., 2020), but this approach is based on reinforcement learning and thus is unstable and time consuming in course of training.

(2) Out of three experimental datasets, two are proposed by the paper for the first time. The authors promise to make them publicly available upon acceptance.

Weaknesses:

(1) The novelty is limited since the proposed approach seems to be only an application of E-LEN. I suggest the authors to strengthen the novelty for clarifying which enhancements are made beyond the framework of E-LEN.

(2) There is no comparison with XGNN in terms of the main metrics namely fidelity, accuracy and concept purity. In particular, the proposed approach seems inferior to XGNN on Mutagenicity in terms accuracy according to the accuracy reported in (Yuan et al., 2020).

(3) The presentation is generally good but can be improved by clarifying the following two questions.

Q1: How to convert a prototype vector (obtained in the concept projection step) to a subgraph (learnt concept) as shown in Figure 3?

Q2: Why not to use the same set of metrics provided in (Yuan et al., 2020) but introduce a new metric concept purity? Besides, concept purity has not been clearly defined in the paper; it should be defined in a more formal way, e.g., by using formulas.


**Summary Of The Paper:**

This paper proposes an approach to applying the Entropy-based Logic Explained Network (E-LEN) to generate global explanations for a GNN from computed local explanations of the GNN, where both local and global explanations are expressed by sub-graphs. Experimental results on three datasets demonstrate good performance in terms of fidelity, accuracy and concept purity.

**Summary Of The Review:**

The paper addresses an important yet insufficiently-studied problem by applying the E-LEN framework. New datasets are proposed to accelerate the on-going studies. However, the proposed approach lacks comparison with the state-of-the-arts in main metrics. Besides, the paper is unclear in some important aspects and need to be improved by clarifying them.

---

### Author Response · Authors · 2022-11-16
**Answer to the reviewers**

We thank the reviewers for their insightful and positive feedback. We are glad they found our work to have a significant impact (R.gQT1), the topic useful (R.CpLz), the paper clear and well organized (R.CpLz), and with a rigorous review of previous works (R.8iKd). We addressed raised questions here and in the revised paper that we uploaded. However, **the core of our contribution and evaluation remains unchanged.**

Our changes can be summarised as:

- We added Appendix A.4, A.5, and A.6 with, respectively, a more detailed description of the metrics, of the E-LEN, and of Eq. 2

- We emphasized the rationale behind our novelty by remarking the differences with the E-LEN framework in A.5.2, and why we could not evaluate XGNN under our metrics in A.2.2

Please, find below some follow-ups to individual comments raised by the reviewers:

---

> ### Author Response · Authors · 2022-11-16
> **Follow-up to the discussion**
>
> **R.gQT1,8iKd - novelty:**
>
> GLGExplainer is composed of simple steps, combined in a way that was not explored before for post-hoc explainability. GLGExplainer key innovations are:
> - Given that the E-LEN requires the space of input concepts to be defined, we devised **a novel unsupervised concept learning mechanism** to convert local explanations in the form of subgraphs into learned concept activations, suitable for formulas extraction with the E-LEN. Note that this conversion is not trivial,
> given that we have no supervision on which concepts to learn and that local explanations may contain spurious nodes/edges
> - **The adaptation of the Entropy Layer** framework, originally described for interpretable-by-design architectures, **to post-hoc settings**
> - **A novel discrete concept activation mechanism** that imposes singleton activation of concepts. This allows the removal of the entropy-based regularization of the Entropy Layer, resulting in an easier-to-be-optimized component (as shown in Fig. 4)
>
> These innovations allowed us to build the first concept-based Global Explainer for GNNs.
> Overall, the fact that our results are showing explanations significantly different (and of higher quality) w.r.t. the existing state of the art, demonstrates that our idea is quite relevant and important, in terms of *usefulness* (as shown in the results), *technical novelty* (as motivated above), and *surprise* (as we are the first to provide global logic explanations for GNNs).
>
>
> **R.8iKd: rationale about GLGExplainer steps:**
>
> Given that our goal is to provide logic explanations of a Neural Network, the state-of-the-art approach for generating such explanations is constituted by the E-LEN. However, the E-LEN framework requires as input a fixed vector of concept activations, which is not readily available from plain local explanations, thus making a direct application of such framework not possible.
> GLGExplainer key steps are:
> - **Extraction of local explanations:** Building the final global explanation on top of local explanations allowed our system to be faithful to the data domain, contrary to the previous state-of-the-art. As mentioned in Section 3 we relied on PGExplainer since it allows the extraction of arbitrary disconnected motifs as explanations and it gave excellent results in our experiments.
> - **Representation learning on local explanations:** This step allows to learn an embedding for each local explanation, represented as a graph. To achieve this, and to keep our system as general as possible, we employed a GIN network, which is provably the most expressive 1-WL GNN based on Message Passing.
> - **Unsupervised concept discovery:** This step allows to automatically learn a set of human-understandable concepts from a set of embeddings. To achieve this, we learned prototypical representations of similar samples, loosely inspired by [2].
>
>
> **R.gQT1 - accuracy compared to XGNN:**
>
> Please note that XGNN was tested on the MUTAG dataset (188 graphs), while instead, we used Mutagenicity (4337 graphs). For this reason, training accuracies (of the GNN to explain *f*) do not match.
> Moreover, both model and local explanations for Mutagenecity are those produced by the original PGExplainer paper (Luo 2020), as mentioned at the end of Section 4.1
>
>
> **R.gQT1, CpLz - comparison with XGNN under our metrics:**
>
> We would like to remark that the explanation provided by XGNN and by GLGExplainer are in two substantially different formats. Thus, **an evaluation of XGNN under our metrics is not effective** for the following reasons:
> - Concept purity: XGNN produces just an explanatory graph, which can not be decomposed nor considered as concepts as intended in [1]
> - Formula Accuracy: XGNN does not produce any logic formulas
> - Fidelity: We defined fidelity as an accuracy-like metric measuring the concordance between our explainer and the GNN to explain. Given that XGNN solely produces as output an explanatory graph, it is not possible to consider this metric
>
> Furthermore, we could not evaluate the class probability as done by XGNN since our logic explanation can not be forwarded to the GNN to be explained.
>
>
> **R.8iKd: the need for a human study evaluating the effectiveness of the explanations:**
>
> For what concerns the evaluation of the explanations, we thoroughly selected the datasets to test on in such a way to enable the validation of the extracted formulas in a controlled setting. This allowed us to relate the formulas of Figure 3 with the underlying composition of each dataset, as described in Section 4.1. Indeed, in Section 4.4 of our paper, **we provide a detailed discussion of our results showing that formulas are giving the expected insights into the model**. For these reasons, we believe that an additional human study is not needed to assess the quality of the explanations.

---

> > ### Author Response · Authors · 2022-11-16
> > **Follow-up to the discussion (2)**
> >
> > **R.gQT1: Q1 - how to convert a prototype vector to a subgraph?:**
> >
> > As described in 4.4 of our paper, Figure 3 represents a more human-readable representation of the formulas of Table 1, where for each concept we chose a representative element among all local explanations assigned to that concept. Figure 2 presents just one local explanation per concept, while Figures 11-12-13 present five random instances for each.
> > For clarity purposes, we slightly updated Figure 3 in order to add the little spurious outgoing edges sometimes present in Figure 2 of BAMultiShapes (in P1, P3, and P4).
> >
> > ---
> >
> > In conclusion, we thank again all the reviewers for the insightful comments, which we believe helped in making our contribution more clear and solid.
> >
> > [1] Human-in-the-Loop Concept-based Explanations for Graph Neural Networks. Magister et al., 2021
> >
> > [2] Deep learning for case-based reasoning through prototypes: A neural network that explains its predictions. Li et al., AAAI 2018

---

### Decision · Program_Chairs · 2023-01-20

**Decision:**

Accept: poster

**Justification For Why Not Higher Score:**

Would like to see higher originality and/or slightly cleaner experimental evaluation to recommend a spotlight.

**Justification For Why Not Lower Score:**

Solid contribution with support for acceptance from all reviewers.

**Metareview: Summary, Strengths And Weaknesses:**

This paper presents a new method for explaining graph neural network predictions, where first local subgraph explanations are computed, then they are combined into a logical formula summarizing the GNN behavior. Reviewers appreciate how the paper brings together ideas to present a compelling approach to a less-studied regime of GNN explanation. Most of the reviewer concerns were addressed in the author responses. While the authors do address the issue in their response and it is reasonable, reviewers still wished that evaluation against SOTA could be made on the metrics from previous work.

**Note From Pc:**

if the above contains the word "oral" or "spotlight" please see: "oral" presentation means -> notable-top-5% and "spotlight" means -> notable-top-25%. As stated in our emails, we are disassociating presentation type from AC recommendations

**Summary Of Ac-Reviewer Meeting:**

The meeting was scheduled, but people were unable to attend at the last minute. However, we had a discussion via email, and the most negative reviewers felt that the author responses addressed their concerns, and they recommended acceptance. At this point, the paper no longer seemed like a borderline paper.